# Evaluating the Predictive Value of Diagnostic Testicular Biopsy for Sperm Retrieval Outcomes in Men with Non-Obstructive Azoospermia

**DOI:** 10.3390/jpm13091362

**Published:** 2023-09-07

**Authors:** Aris Kaltsas, Eleftheria Markou, Athanasios Zachariou, Fotios Dimitriadis, Evangelos N. Symeonidis, Athanasios Zikopoulos, Charalampos Mamoulakis, Dung Mai Ba Tien, Atsushi Takenaka, Nikolaos Sofikitis

**Affiliations:** 1Department of Urology, Faculty of Medicine, School of Health Sciences, University of Ioannina, 45110 Ioannina, Greece; a.kaltsas@uoi.gr (A.K.); kzikop@uoi.gr (A.Z.); 2Department of Microbiology, University Hospital of Ioannina, 45500 Ioannina, Greece; eleftheria.markou4@gmail.com; 3Department of Urology, Faculty of Medicine, School of Health Sciences, Aristotle University of Thessaloniki, 54124 Thessaloniki, Greece; difotios@auth.gr (F.D.); evansimeonidis@gmail.com (E.N.S.); 4Department of Urology, Faculty of Medicine, School of Health Sciences, University of Crete, 70013 Heraklion, Greece; mamoulak@uoc.gr; 5Department of Andrology, Binh Dan Hospital, Ho Chi Minh City 70000, Vietnam; maibatiendung@yahoo.com; 6Division of Urology, Department of Surgery, School of Medicine, Faculty of Medicine, Tottori University, Yonago 683-8503, Japan; atake@med.tottori-u.ac.jp

**Keywords:** diagnostic testicular biopsy, therapeutic testicular biopsy, non-obstructive azoospermia, sperm retrieval, microdissection testicular sperm extraction

## Abstract

Background: Non-obstructive azoospermia (NOA) presents a challenge in male infertility management. This study aimed to assess the efficacy of diagnostic testicular biopsy (DTB) in predicting sperm retrieval success via therapeutic testicular biopsy (TTB) and to understand the role of systemic inflammation in microdissection testicular sperm extraction (mTESE) outcomes. Methods: A retrospective analysis was conducted on 50 NOA males who underwent mTESE at the University of Ioannina’s Department of Urology from January 2017 to December 2019. All participants underwent thorough medical evaluations, including semen analyses and endocrinological assessments. Results: DTB did not detect spermatozoa in half of the patients who later showed positive sperm findings in TTB. Preoperative variables, such as age, plasma levels of follicle-stimulating hormone (FSH), luteinizing hormone (LH), total testosterone (TT), prolactin (PRL), estradiol (E2), and inflammation biomarkers (neutrophil–lymphocyte ratio (NLR), platelet–lymphocyte ratio (PLR), monocyte–eosinophil ratio (MER)), were not consistently predictive of sperm retrieval success. Notably, TTB-negative patients had elevated NLR and PLR values, suggesting a possible link between systemic inflammation and reduced sperm retrieval during mTESE. Conclusions: The findings question the necessity of an initial DTB, which might provide misleading results. A negative DTB should not deter further TTB or intracytoplasmic sperm injection (ICSI) attempts. The study emphasizes the need for further research to refine diagnostic approaches and deepen the understanding of factors influencing sperm retrieval in NOA patients, ultimately enhancing their prospects of biological parenthood.

## 1. Introduction

Non-obstructive azoospermia (NOA) is a critical form of male infertility distinguished by a deficit of spermatogenesis within seminiferous tubules, impacting approximately 10–15% of infertile men and constituting the leading cause of male infertility, corresponding to 60% of all cases [1]. Although azoospermia suggests the absence of spermatozoa in the ejaculate, a potential exists for sperm retrieval from the testicles, provided spermatogenesis is still active [1]. The combined approach of intracytoplasmic sperm injection (ICSI) and testicular sperm extraction (TESE) offers the potential for men with NOA to father biological offspring [2]. Diagnostic testicular biopsy (DTB) is a diagnostic procedure employed to ascertain the cause of azoospermia, to rule out the presence of neoplasia, and to assess disruptions in spermatogenesis by analyzing histological sections of testicular tissue. Conversely, therapeutic testicular biopsy (TTB) is a therapeutic procedure focused on extracting sperm from a more extensive portion of testicular tissue for reproductive applications. Microdissection testicular sperm extraction (mTESE) has emerged as a pivotal tool in this endeavor, markedly enhancing the testicular sperm retrieval rates (SRRs) for men with NOA compared to conventional procedures [2]. Nonetheless, the success of sperm retrieval in these patients can be influenced by a myriad of factors, mainly systemic inflammation, and the role of DTB, which is the primary focus of this study [3,4,5].

While NOA can be linked to various medical conditions, such as cryptorchidism, varicocele, hypogonadotropic hypogonadism, chromosomal defects, and Klinefelter’s syndrome, systemic inflammation and chronic diseases such as atherosclerosis and cancer are particularly influential due to their damaging impact on sperm quality [6]. Oxidative stress from these inflammatory processes has been associated with sperm DNA and membrane damage, potentially resulting in azoospermia [6]. Moreover, in nearly 20% of azoospermic infertile individuals, testicular biopsies show immune cell infiltration, indicating the possible underestimation of inflammatory infertility’s significance [7]. Consequently, the need arises to delve into the impact of systemic inflammation on SRR in NOA patients.

Moreover, inflammatory markers, with recent attention given to the neutrophil-to-lymphocyte ratio (NLR), monocyte-to-eosinophil ratio (MER), and platelet-to-lymphocyte ratio (PLR), have been linked to several diseases, including cardiovascular disease [8]. Exploring the connection between these markers and SRR in NOA patients might help decipher the intricate relationship between inflammation and reproductive outcomes.

In this context, testicular biopsy serves as a crucial diagnostic tool, distinguishing between obstructive and non-obstructive azoospermia and predicting mature spermatid recovery through mTESE. [7]. While this tool is essential, varying histological reporting methods and unclear terms can reduce the reliability of studies on mTESE recovery rates [7]. Further, the high prevalence of testis cancer and carcinoma in situ (CIS) in infertile populations necessitates early detection, advocating for a systematic approach to the histological classification of spermatogenic disorders and CIS detection in adult patients [7].

Surgical sperm retrieval in azoospermic patients can be affected by several factors, including age, testicular volume, reproductive hormone levels, and preoperative DTB [9]. However, not all these factors have been found to significantly impact sperm acquisition in patients undergoing mTESE [9]. In particular, despite its crucial role in predicting mTESE success in men with NOA, DTB does not always significantly influence sperm retrieval rates [10]. The presence of mature spermatozoa in a histopathology specimen is a robust predictor of successful TESE. However, conducting a diagnostic biopsy before mTESE is not always practical due to the requirement for two surgeries [10]. As such, it is often more feasible to couple the diagnostic biopsy with an initial mTESE before starting the ICSI cycle [10].

A key observation from our study, and one that aligns with the broader literature, is the heterogeneous nature of spermatogenesis within the testes of NOA patients. This heterogeneity underscores the importance of our findings regarding the limited predictive value of DTB. While a positive DTB result can be indicative of successful sperm retrieval, a negative result does not necessarily preclude the presence of sperm during mTESE. This observation is crucial for clinicians as it suggests that relying solely on DTB results might lead to missed opportunities for sperm retrieval. Our study emphasizes the need for a more nuanced approach, considering both the DTB results and other predictive factors, to optimize the chances of successful sperm retrieval in NOA patients.

Lastly, the role of hormonal treatment, encompassing factors such as follicle-stimulating hormone (FSH), luteinizing hormone (LH), and total testosterone (TT) in maximizing sperm retrieval yield in NOA patients has been explored [11]. Despite conflicting results in the efficacy of such treatments prior to mTESE, they represent a potential approach to improve SRR in NOA patients and thus warrant further research [11].

Given the complex landscape of factors influencing sperm retrieval in men with NOA, this retrospective study aimed to assess the impact of systemic inflammation and DTB on sperm retrieval rates in NOA. This study also evaluated the predictive values of factors such as reproductive hormones, testicular histopathology, testicular volume, and male age to better inform decision making in cases requiring reintervention. Notably, despite the indicators discussed, achieving absolute certainty in predicting the presence of sperm at mTESE before surgery remains a challenge, underscoring the need for further research.

## 2. Materials and Methods

### 2.1. Study Design and Participants

Fifty males with NOA were included in this retrospective investigation. All patients underwent mTESE at the Department of Urology of the University of Ioannina between January 2017 and December 2019. Clinical assessment (including palpation of vas deferens bilaterally) of infertile men was performed. Comprehensive medical histories, physical examinations, semen analyses, and endocrinological evaluations were performed on all patients. Patients were diagnosed with azoospermia when at least two semen studies using centrifugation and thorough pellet inspection showed no spermatozoa, the ejaculate had a more than 1.0 mL volume, and its pH was higher than 7.2. 

The current study was performed in the Laboratory of Spermatology of our department. In this laboratory, oligozoospermic, asthenozoospermic, teratozoospermic men, and azoospermic men, represent the majority of participants. After performing semen analysis/semen sample centrifugation, each azoospermic man was further analyzed. 

We performed mTESE on the 50 men who had been considered pre-mTESE as NOA-men using the following examinations:○Studying medical and sexual history, physical examination results; ○Evaluating peripheral serum hormonal levels (FSH, LH, TT, estradiol (E2), prolactin (PRL)), testicular volume (TV);○Performing scrotal ultrasonography, transrectal ultrasonography, and karyotype testing;○Evaluating for Y-chromosomal microdeletions. 

No males in the current study had been given exogenous testosterone or any other medication.

The normal ranges for hormonal levels are determined by the standard values of the equipment used in our laboratory. Males with high FSH or small testicular volume/atrophic testes were considered NOA-men [12]. To distinguish between obstructive azoospermia and NOA in males with normal FSH and normal testicular volume, we were assisted by the findings of the physical examination (i.e., presence or absence of vas deferens, presence or absence of dilated epididymis, among others), the transrectal ultrasound, the scrotal ultrasound, the karyotype, and the Y-chromosome microdeletions outcome. According to the European Association of Urology Guidelines on Sexual and Reproductive Health 2023 [12], the DTB and TTB should be performed during mTESE. 

The present study is one of the few studies in the international literature that evaluates three inflammatory indexes for predicting the outcome of mTESE. The NLR was calculated by dividing the total neutrophil count by the total lymphocyte count. The PLR was calculated by dividing the total platelet count by the total lymphocyte count, and the MER was calculated by dividing the total monocyte count by the total eosinophil count. Normal ranges for these cell counts, as determined by the standard values of the equipment used in our laboratory, are as follows: leukocyte 4–11 × 10^3^/μL; neutrophil 2–7.5 × 10^3^/μL; lymphocyte 1–4.5 × 10^3^/μL; monocyte 0.2–0.8 × 10^3^/μL; and platelet 150.0–400.0 × 10^3^/μL (data from our laboratory). Exclusion criteria included obstructive azoospermia, recognized cytogenetic abnormalities, Y-chromosome microdeletions, and testosterone levels below 300 ng/mL.

Only patients who granted informed permission for the surgical operation and the anonymous use of their clinical data were included in the research. All men underwent mTESE, and a small section of testicular tissue was sent for histopathological examination (DTB). An additional piece of testicular tissue was sent for processing by a specialist embryologist to find spermatozoa (TTB). The remaining and more significant portion of the testicular tissue was sent for cryopreservation. The sperm found in TTB were also forwarded for cryopreservation in assisted reproduction techniques for future use. DTB results were compared with TTB. 

### 2.2. Technique of mTESE

The mTESE procedure was conducted under general anesthesia by a consistent surgical team, which included Professor Nikolaos Sofikitis as the first surgeon and Aris Kaltsas as the assistant surgeon (Figure 1). Both the scrotal skin and all testicular coverings were incised. We examined the testicular parenchyma under an operating microscope through a longitudinal opening in the tunica albuginea. We focused on recovering opaque and dilated seminiferous tubuli with larger diameters from each patient. The microscope used was a LEICA M 651 model (Figure 1).

The mTESE was performed bilaterally. For each participant, we obtained four to ten micro-samples from each testis. A small portion of the recovered testicular tissue was set aside for histopathological examination, a DTB. As previously described, the remaining testicular tissue samples were prepared and processed for cryopreservation [13]. 

We decided not to employ an embryologist in the operating theater to mince the testicular tissue and search for haploid gametes. This was primarily because an extensive period of testicular tissue mincing is required in a select subpopulation of NOA-men (i.e., men with non-mosaic Klinefelter’s syndrome). In some cases, enzymatic digestion is necessary [14]. Consequently, the presence of an embryologist in the operating theater would substantially extend the duration of the mTESE procedure.

### 2.3. Testicular Histopathology 

All testes’ samples were sent to the same pathologist for histological analysis. A small piece of subcapsular parenchyma was taken, preserved in Bouin’s solution, and sent to the pathologist for analysis. Hematoxylin and eosin (H & E) were used to stain histologic slides. At least a hundred tubuli seminiferous slices were analyzed histologically. Moreover, testicular intraepithelial neoplasia and other abnormalities were examined histopathologically. Results from the histology analysis were classified as hypospermatogenesis (HYPO), characterized by a severely reduced population of germ cells in the tubules; all stages of germ cells (spermatogonia, spermatocytes, and spermatids) were present but in reduced numbers. Early and late maturation arrest (MA) is characterized by an arrest of the spermatogenetic maturation sequence at the primary spermatocyte and round spermatid levels, respectively. Sertoli cell-only syndrome (SCO) is a condition where the tubules are predominantly or exclusively populated by only Sertoli cells.

### 2.4. Processing of Therapeutic Testicular Biopsy Material

The TTB was delivered to the spermatology laboratory, where it was mechanically and enzymatically minced. A confocal laser scanning microscope and computer system were used to analyze most of the fragmented testicular tissue (CLSM-CAS [15]). Samples from therapeutic testicular biopsies were cleaned four times in normal saline. Finally, the seminiferous tubules were cut into tiny pieces and washed with Dulbecco’s phosphate-buffered saline (DPBS; Sigma Co., St Louis, MO, USA) containing 5.6 mmol/L glucose and 5.8 mmol/L sodium lactate (modified DPBS [16]). Samples were kept at 5 °C Celsius and viewed under a dissecting microscope (Olympus SZ-STS; Olympus, Tokyo, Japan) to monitor the comminution process. After the samples were centrifuged at 500 g for 30 min, the sedimented tissue and cell fragments were suspended in modified DPBS and filtered through paper with a 30–40 m pore size. The sedimented tissue and cell fragments were then removed from the paper. After collecting the filtrate and centrifuged at 750 g for 30 min, the majority (main fraction) of the sedimented cells were analyzed using a CLSM-CAS. The presence or absence of sperm was used to determine whether the result was TTB positive or negative.

### 2.5. Ethical Approval 

The study protocol, involving the analysis of previously collected data, received approval from the departmental ethics committee, ensuring alignment with the guidelines of the Helsinki Declaration. While the data was collected for prior clinical purposes, all patient information included in this study was fully anonymized to protect patient confidentiality. Any identifiable information was excluded or de-identified before analysis to maintain the privacy and integrity of the patient data.

### 2.6. Statistical Analysis

In our statistical analysis, we initially employed *t*-tests to assess differences in blood hormone levels, inflammatory biomarkers, and testicular histopathology between patients stratified by SRR. While *t*-tests provided preliminary insights into these differences, we also utilized chi-square (χ2) tests to analyze variations in testicular histopathology among the stratified groups. The sample size of 50 was determined based on practical constraints, including the availability of participants during the study period and budgetary considerations. While we acknowledge that a larger sample size might have provided more robust findings, the chosen sample size was deemed the most feasible given the constraints. 

Recognizing the complexities of our data, especially with multiple covariates, we further employed binary logistic regression, a form of General Linear Model (GLM), to delve deeper. SRR was used as the dependent variable in a binary logistic regression model, with hormone levels and inflammatory biomarkers as continuous variables, and testicular histology as a categorical (nominal) variable using contrast coding. This model allowed us to comprehensively analyze the relationships and account for the multiple covariates. 

The predictive accuracy of each variable was independently measured using the area under the curve (AUC) of the receiver operating characteristic (ROC) evaluations. We then used binary logistic regression to assess the diagnostic accuracy of our prediction model, specifying the presence or absence of sperm in men as a binary dependent variable. The AUC of the ROC quantified the predictive value of all variables assessed individually. SRR in patients stratified by testicular histology was computed via χ2 analysis. 

Regarding the assumptions of the *t*-test, we acknowledge the importance of data normality and homoscedasticity. Prior to conducting the *t*-tests, we assessed these assumptions using Shapiro–Wilk tests for normality and Levene’s test for equality of variances.

Given the binary nature of our primary outcomes and the specific comparisons we aimed to make, we believe our combined approach of *t*-tests and binary logistic regression was appropriate for our study’s objectives. All statistical analyses were conducted using IBM SPSS Statistics 26 and a *p*-value of less than 0.05 was considered statistically significant.

Despite our rigorous approach, it is important to note that our results are exploratory in nature and should be interpreted with caution. We recommend larger, future studies with more advanced statistical methods to validate and extend our findings. 

## 3. Results

We present the findings from our retrospective analysis of 50 patients with NOA, detailing their demographic characteristics, underlying causes of NOA, and the correlation between various diagnostic parameters and the success of sperm retrieval. In total, 50 patients who met the criteria for inclusion in the current study were enrolled. The mean age of this group was 38.5 years, with a standard deviation of 7.1 years and a range from 19 to 61 years (Figure 2a). 

The various causes of NOA were investigated using the collected clinical data. Clinical left varicocele was diagnosed in nine (18%) men. Among these, four men had varicocele grade 2, three had varicocele grade 3, and two had varicocele grade 1 [17]. It is noteworthy to mention that while varicocele, especially of higher grades, has been associated with NOA, the presence of grade 1 varicocele in two patients raises questions about its sole contribution to NOA. Four (8%) NOA patients were diagnosed with Klinefelter’s syndrome, five (10%) NOA patients with urogenital tract infection history, six (12%) men with cryptorchidism, and two with testicular torsion. In our cohort, none of the patients exhibited Y-chromosome microdeletion, a finding that diverges from the established literature. Twenty-four (48%) of the NOA cases had no identifiable cause and were thus classified as idiopathic (Figure 2b). 

A bilateral approach was performed in 96% of patients because three had previously had an orchiectomy. No significant complications (>grade 1 Clavien–Dindo [18]) occurred during or after surgery. DTB was performed in all patients at the same time as TTB. In our comprehensive study of patients undergoing TTB, we also evaluated the histological patterns observed in DTB. This parallel assessment aimed to understand the correlation between the histological findings in DTB and the outcomes of TTB. Table 1 provides a detailed breakdown of the histological patterns observed in DTB and their corresponding TTB outcomes. By analyzing these patterns, we aim to discern any potential predictive value of DTB histology in determining TTB success.

From the data presented in Table 1, it is evident that the histological patterns observed in DTB have varying associations with TTB outcomes. Notably, all patients with a HYPO pattern in DTB had a positive TTB result, suggesting a strong correlation between HYPO histology and successful sperm retrieval. Conversely, the majority of patients with SCOS in DTB had negative TTB outcomes, indicating a potential challenge in sperm retrieval for this group. The MA pattern showed a more balanced distribution between positive and negative TTB outcomes. Overall, the TTB-positive rate among the NOA-men was 44%, while the TTB-negative rate stood at 56%. Interestingly, while DTB reported spermatozoa in 11 patients, TTB found spermatozoa in double that number, highlighting the potential differences in the diagnostic and therapeutic approaches. This analysis underscores the importance of understanding the histological patterns in DTB as potential predictors for TTB outcomes, although further research might be necessary to solidify these findings.

The subsequent table, Table 2, provides a comprehensive overview of the preoperative peripheral hormonal serum values and blood biomarkers of systemic inflammation in a cohort of 50 men. These values are further stratified based on the success of sperm retrieval, categorized as either TTB-negative or TTB-positive. By comparing the mean values and standard deviations of each variable between the two groups, the table aims to elucidate potential correlations and significant differences that might influence the outcome of sperm retrieval.

From the data presented in Table 2, it is evident that certain hormonal and inflammatory markers exhibit significant differences between men who tested positive for sperm retrieval and those who did not. Specifically, FSH levels were notably lower in men with successful sperm retrieval, suggesting its potential role as a predictive marker. On the other hand, while LH levels were also lower in the successful group, the difference was not statistically significant. In terms of inflammatory biomarkers, both NLR and PLR values were elevated in men who tested negative for sperm retrieval, hinting at a possible association between systemic inflammation and reduced sperm retrieval success. However, other markers such as TT, PRL, E2, and MER did not show significant variations between the groups. Overall, these findings underscore the importance of considering both hormonal and inflammatory profiles in predicting the success of sperm retrieval procedures.

In our pursuit to understand the factors influencing successful TTB sperm retrieval, we employed a multiple logistic regression analysis. This statistical approach allows us to assess the individual and combined effects of various variables on the likelihood of successful sperm retrieval. Table 3 presents the results of this analysis, detailing the coefficients, standard errors, Wald’s statistics, *p*-values, odds ratios, and confidence intervals for each variable. By examining these metrics, we aim to identify key predictors that could enhance the diagnostic accuracy of our prediction model for successful sperm retrieval.

Among the variables analyzed in Table 3, DTB stands out as having a significant association with successful sperm retrieval, although the wide confidence interval suggests considerable uncertainty around the magnitude of this effect. FSH also shows a potential association, but it is borderline significant. The other variables do not show statistically significant associations with successful sperm retrieval in this analysis. The variable DTB, given its significance, might be of clinical importance in predicting successful sperm retrieval, but further studies might be needed to narrow down the confidence interval and provide a more precise estimate.

Beyond the individual variable analysis, it is crucial to understand the collective performance of the predictors in the model. To evaluate the overall significance and explanatory power of the regression model, we conducted an analysis of variance (ANOVA) and summarized the model’s performance metrics. Table 4 presents the ANOVA results, which test the overall fit of the model, while Table 5 provides a summary of the regression model, detailing the correlation and variance explained by the predictors.

The ANOVA results from Table 4 indicate that the overall model, while encompassing a range of predictors, does not significantly predict TTB, as evidenced by the *p*-value of 0.333. Furthermore, the model summary in Table 5 reveals that the set of predictors explains approximately 23.2% of the variance in TTB, but after adjusting for the number of predictors, this explanatory power drops to a mere 3.5%. This underscores the complexity of predicting TTB and suggests that while some individual predictors might have significance, their collective power in the current model is limited. Future research might benefit from exploring additional variables, refining the model, or employing more advanced modeling techniques to enhance predictive accuracy.

To better understand the predictive capabilities of various variables in determining the success of sperm retrieval, we employed the ROC curve analysis. This statistical method evaluates the diagnostic ability of a binary classifier system, with the AUC serving as a key metric to quantify the accuracy of each predictor. Figure 3 presents the ROC curve analysis for each variable, offering insights into their individual diagnostic accuracies in predicting positive or negative sperm retrieval results.

The ROC curve analysis, as depicted in Figure 3, provides a comprehensive assessment of the predictive power of each variable. Age and hormonal tests, including FSH, LH, TT, PRL, and E2, exhibited AUC values below 0.7, indicating their limited efficacy in accurately predicting sperm retrieval outcomes. TV’s AUC of 0.678 suggests its marginal predictive power for mTESE success. In contrast, inflammatory biomarkers MER, NLR, and PLR demonstrated even lower AUC values, further emphasizing their limited diagnostic accuracy. However, the standout predictor was DTB, with an AUC of 0.892, highlighting its strong predictive capability. This suggests that DTB, among the variables studied, offers the most reliable diagnostic accuracy in forecasting the success of sperm retrieval. The findings underscore the importance of considering multiple factors in tandem rather than relying on singular predictors when evaluating the likelihood of successful sperm retrieval in NOA patients.

The model, constructed using all pertinent variables, underwent an ROC curve analysis. Figure 4 illustrates the results of this analysis, providing a visual representation of the model’s predictive capability.

From Figure 4, the derived AUC value stands at 0.216. This suggests a constrained ability of the variables to reliably forecast sperm retrieval outcomes. Typically, AUC values under 0.7 indicate weak predictive capacity.

In our continued analysis, we further evaluated the predictive capability of our model using the ROC curve analysis. Table 6 presents the detailed results, including the AUC value, which serves as a crucial metric to gauge the model’s overall discriminatory power.

The AUC value of 0.216, as detailed in Table 6, suggests that the model has limited predictive power. Typically, an AUC value below 0.7 indicates limited to no predictive capability. The low AUC value emphasizes the challenges in predicting sperm retrieval outcomes using the current set of variables.

Beyond assessing the predictive power of the model, it is crucial to evaluate the level of agreement between the observed outcomes and the predictions made by the model. Cohen’s kappa is a statistic that measures this agreement for categorical items, adjusting for what might be expected by chance. Table 7 provides the results of the Cohen’s kappa evaluation for our study.

The Cohen’s kappa value of 0.000, presented in Table 7, indicates no agreement between the observed TTB outcomes and the predictions based on the significant variables in our model. This value suggests that the model’s predictions align with the actual outcomes no better than what would be expected by random chance. Such a result underscores the need for refining the predictive model or considering additional variables to enhance its accuracy.

In the realm of medical diagnostics, the ability of a test to correctly identify and classify cases is of paramount importance. Sensitivity, specificity, positive predictive value (PPV), and negative predictive value (NPV) are critical metrics that provide insights into the performance of a diagnostic test. In the context of our study, we evaluated the DTB to determine its efficacy in predicting successful sperm retrieval in TTB. The results, summarized in Table 8, shed light on the diagnostic accuracy of DTB in this clinical scenario.

The data presented in Table 8 underscore the robust diagnostic capability of the DTB. With a specificity of 100%, the DTB demonstrates an impeccable ability to correctly identify all true negative cases. This high specificity, coupled with a commendable positive predictive value of 100%, suggests that when DTB indicates a positive result, there is a very high likelihood of successful sperm retrieval in TTB. However, with a sensitivity of 50%, there’s room for improvement in detecting true positive cases. Overall, while DTB showcases significant promise as a predictor for successful sperm retrieval in TTB, clinicians should interpret the results in conjunction with other clinical parameters and patient-specific factors.

In summary, our results indicate that while hormonal levels and inflammatory markers offer some insight, DTB stands out as the most potent predictor for successful sperm retrieval in TTB. The histopathological findings from DTB were the only significant contributors to the logistic regression model, highlighting the critical role of histopathological examination in the diagnosis and management of patients with NOA.

## 4. Discussion

Infertility is a medical issue that has impacted couples for many years, and our understanding and management of this issue have continued to evolve. Before 1995, the options available to couples dealing with NOA were limited; in vitro, conception with donor sperm, or adoption, were the main paths to parenthood [19,20]. However, technological advancements and medical discoveries have expanded the possibilities for couples facing these challenges [21].

The development of TTB allowed clinicians to retrieve sperm directly from the testes of NOA patients, opening a new avenue for these couples to have biological offspring through ICSI. The combined approach of TTB and ICSI became a successful therapeutic option to manage male infertility due to NOA. The current gold standard in this realm combines mTESE with ICSI cycles [22,23]. Using a surgical microscope for examining seminiferous tubules is crucial for the success of this procedure, as it improves the odds of successful sperm retrieval and minimizes the risk of injury to the testicles, especially compared to the earlier conventional TESE methods [24]. However, despite these advances, the sperm recovery rate via mTESE, as per the literature review, still ranges between 40% and 60%—a figure that is less than satisfactory when considering this procedure’s physical, emotional, and financial implications [23].

Our research explored potential predictive factors of successful sperm retrieval using TTB in NOA patients. An important focus of our study was to investigate the impact of various factors, such as reproductive hormones, inflammatory biomarkers, biopsy histology, testicular volume, and male age, on sperm retrieval success. One major finding of our study was the significant predictive value of testicular histopathology in the outcome of mTESE. This result concurs with previous research showing the importance of DTB. The presence of tubules with mature spermatozoa on biopsy was the best predictor of positive surgical sperm retrieval. This aligns with previous studies highlighting the importance of DTB in identifying the underlying causes of male infertility and differentiating between primary testicular damage and obstructions in the reproductive tract [10,25].

An intriguing observation in our cohort was the absence of Y-chromosome microdeletion among the participants. Historically, deletions within the male-specific region of the Y-chromosome, known as Y-chromosome microdeletions (YCMs), have been identified in a significant proportion of azoospermic men. The prevalence of these microdeletions varies based on region and ethnicity, with some studies reporting their presence in up to 10% of azoospermic men [26]. The absence of this microdeletion in our study raises questions about the genetic landscape of our cohort and suggests potential regional or demographic variations. It is essential to recognize that while YCMs are a recognized factor in NOA, their absence in our study underscores the multifactorial nature of male infertility and the need for comprehensive genetic evaluations. This finding also emphasizes the importance of considering other genetic, environmental, and physiological factors that might contribute to NOA in specific populations.

A significant percentage of men with testicular histology of SCOS or spermatogenic arrest at the primary spermatocyte stage on DTB have testicular foci of active spermatogenesis to the secondary spermatocyte stage, elongating spermatids, round spermatids, or spermatozoa, which calls into question the importance of DTB in the therapeutic management of non-obstructed azoospermic men [27,28,29]. Therefore, a DTB cannot identify SCOS men who test positive for testicular foci containing haploid cells and are potential candidates for assisted reproduction programs. The presence of foci of haploid cells in therapeutic testicular biopsies from SCOS men and typically nondisabled azoospermic men cannot be reliably predicted by peripheral blood concentrations of FSH or testicular size alone [29].

A DTB was conducted in all patients concurrently with TTB to determine the extent of testicular damage and rule out potential intratubular germ cell neoplasia of an unclassified type (ITGCNU), occurring in 1–5% of infertile men [30]. The testicular tissue sample for DTB was placed in Bouin’s solution before being sent for histological examination with hematoxylin–eosin staining [31]. The investigation concluded with an immunohistochemical test to eliminate the possibility of testicular neoplasia [31]. None of the men tested positive for ITGCNU or any other form of testicular germ cell neoplasia. Spermatozoa were visible in the histological preparation of DTB in 11 patients. The testicular tissue sample used for TTB was initially subjected to mechanical dissection and enzymatic digestion, leading to the discovery of spermatozoa in 22 patients. Notably, the success rate of sperm identification was statistically significantly higher with TTB than with DTB. Sperm was successfully found in TTB in 100% of men with HYPO based on DTB, 40% with MA, and 14% with SCOS. Among the three histological patterns observed, men with the SCOS pattern in DTB had a lower sperm retrieval rate compared to men with HYPO and MA.

The predictive value of DTB for TTB in sperm-positive men was good (AUC = 0.892). The rate of sperm retrieval was significantly higher in HYPO patients compared to SCOS or MA patients (100% vs. 7% and 40%, respectively, *p* < 0.05). The model was statistically significant (*p* < 0.05), correctly classifying 78% of cases with a sensitivity of 50% (95% CI 38.22–71.78) and a specificity of 100% (95% CI 87.66–100), PPV of 100%, and NPV of 78%. Interestingly, in the DTB results, no sperm were found in 11 patients, whereas in the corresponding TTB samples from the same patients, sperm was detected. This discrepancy may stem from the differing scientific interpretations by pathologists of the same testicular tissue sample. Indeed, a testicular tissue sample following a biopsy may not accurately reflect the dynamic spermatogenesis occurring in the entire testicular tissue. It is worth noting that spermatogenesis in some testes is inherently heterogeneous, which means that a negative DTB result does not necessarily indicate the failure of sperm retrieval. As mentioned in the literature, spermatogenesis in NOA patients’ testes can be focal [32]. Additionally, the TTB processing of testicular tissue, which involves mechanical disruption and enzymatic digestion, seems to enhance the likelihood of discovering spermatozoa. Due to its limited predictive value, the European Association of Urology guidelines do not recommend performing DTB before TTB [12]. This is because the DTB result is seen as insufficient for identifying those men with NOA who would be TTB sperm-positive and candidates for ICSI. Our study also found that nearly half of the patients with NOA, whose spermatozoa were not identified in DTB, were eventually found to have spermatozoa in TTB. This suggests that while DTB can provide some indication, it does not definitively identify men with testicular spermatozoa. It is important to note that a negative DTB result should not deter patients from attempting future TTB and subsequent ICSI cycles, as DTB may not accurately identify men who are negative for spermatozoa in TTB. However, for patients who previously had unsuccessful TTB sperm retrieval attempts, the DTB result could offer valuable information for selecting the appropriate surgical technique for sperm retrieval in subsequent TTB attempts. For instance, in a subpopulation of patients with failed sperm retrieval surgery, those exhibiting a HYPO histological pattern in DTB might consider another TESE attempt rather than the more invasive, time-consuming mTESE surgery requiring specialized personnel and microscope usage. On the other hand, in more severe cases, such as patients displaying a SCOS histological pattern in DTB, the only recognized method of sperm retrieval is mTESE. The success rates of mTESE in NOA patients needs further investigation.

Age was also investigated as a potential factor influencing the success of sperm retrieval in our study. We aimed to address the gap in knowledge regarding the impact of paternal age on the outcomes of mTESE in men with NOA. However, our findings did not demonstrate a statistically significant correlation between age and sperm retrieval success [33]. Further research is needed to fully understand the relationship between age and sperm retrieval rates in NOA patients. While our study did not find a statistically significant correlation between paternal age and sperm retrieval success, it is worth noting that the broader literature suggests a rise in paternal age correlates with decreased sperm quality, compromised testicular function, and potential adverse reproductive outcomes, emphasizing the importance of informed guidance for infertile couples regarding the potential risks associated with advanced paternal age [34].

Reproductive hormone levels, including FSH, LH, TT, PRL, and E2, were also examined as potential predictors of sperm retrieval success. While there were variations observed in hormonal markers between groups categorized by positive and negative sperm retrieval outcomes, these differences did not demonstrate statistical significance as predictors. This aligns with previous research indicating significant interindividual variability in hormonal levels, which may not directly correlate with testicular function or sperm production [35,36].

Inflammatory markers, such as NLR, PRL, and MER, were also evaluated in our study. Elevated levels of these markers were observed in men who yielded negative results for sperm retrieval, indicating a potential association between systemic inflammation and non-obstructive azoospermia NOA. However, further research is needed to establish the predictive value of systemic inflammation in sperm retrieval success [5].

In our pursuit to understand the predictors of successful sperm retrieval in NOA patients, our study embarked on a comprehensive analysis of various influential factors. We utilized a regression model to assess the collective efficacy of these predictors.

Table 4’s ANOVA results, aimed at testing the regression model’s fit, reveal that despite the inclusion of a diverse range of predictors, the model does not significantly predict TTB, as highlighted by the *p*-value of 0.333. This indicates that the collective influence of the predictors does not substantially account for the variance in TTB outcomes.

Table 5 delves into the performance metrics of the regression model. The predictors collectively account for roughly 23.2% of the variance in TTB. Yet, when adjusted for the number of predictors, this explanatory capacity dwindles to just 3.5%. This highlights the inherent challenges in predicting TTB. While certain predictors may individually hold significance, their combined efficacy in this model appears limited.

Figure 4 offers a visual insight into the ROC curve analysis, shedding light on the model’s predictive prowess. The AUC value, standing at 0.216, points towards a limited capability of the predictors to consistently forecast sperm retrieval outcomes. As a general benchmark, AUC values below 0.7 are indicative of subpar predictive strength.

Table 6 further elaborates on the ROC curve analysis results. The AUC value of 0.216 reiterates the model’s constrained predictive capacity. This value accentuates the inherent difficulties in forecasting sperm retrieval outcomes using the current predictors.

Furthermore, it is imperative to gauge the congruence between the model’s predictions and the actual observed outcomes. Table 7’s Cohen’s kappa evaluation reveals a stark absence of agreement between the observed TTB outcomes and the model’s predictions. A Cohen’s kappa value of 0.000 suggests that the predictions resonate with the actual outcomes merely by chance. Such findings emphasize the pressing need to either refine the current predictive model or to incorporate additional predictors to bolster its accuracy.

Given these insights, there is a clear avenue for future research to delve into other potential predictors, fine-tune the existing model, or leverage more sophisticated modeling techniques to bolster predictive precision. Given the intricate nature of sperm retrieval in NOA patients, a comprehensive approach that weighs both individual and combined predictors is paramount to craft a more reliable prediction model.

In our study, we observed an association between elevated NLR and PLR values and negative sperm retrieval outcomes, suggesting a potential clinical relevance of these markers in the context of male infertility. However, it is essential to approach these findings with caution. When subjected to ROC analysis, the AUC values for both NLR and PLR were not sufficiently high, underscoring their limited diagnostic accuracy. While these markers may provide some insights into sperm retrieval outcomes, their AUC values suggest that they might not serve as the most reliable standalone predictors. Therefore, relying solely on NLR and PLR values for diagnostic purposes could be misleading, and it would be prudent to consider them in conjunction with other diagnostic tools and clinical evaluations.

Varicocele, a condition characterized by the dilation of veins in the scrotum, has been implicated in male reproductive dysfunction [37]. While our study identified varicocele in several NOA patients, the association between varicocele grade 1 and NOA is intriguing. Recent studies have emphasized the significance of correctly classifying clinical varicocele, as the grade of varicocele has been linked to treatment prognosis. Specifically, high-grade varicoceles are more frequently associated with azoospermia, while varicocele grade I might not be the sole explanation for the entire disease etiology [38,39,40]. This is further corroborated by findings suggesting that in men with NOA and varicocele, the treatment response, indicated by the presence of sperm in the ejaculate post-varicocelectomy, was more favorable in patients with higher grade and bilateral varicocele. Therefore, while varicocele, especially of higher grades, can be a contributing factor to NOA, it is essential to consider other potential etiological factors, especially in cases of grade 1 varicocele [40]. Interestingly, our findings did not show a significant impact of varicocele on the success of sperm retrieval in men with NOA. This is consistent with previous research that has shown inconsistent results regarding the effect of varicocele on sperm retrieval rates [41]. Despite the debate regarding the benefits of varicocelectomy in patients with NOA, some studies suggest that its repair may contribute to the reappearance of spermatozoa in semen, potentially improve testicular sperm recovery rate, enhance pregnancy rates in ICSI programs, and reduce the need for testosterone replacement therapy in cases of late-onset hypogonadism [40].

Limitations: Although the retrospective nature of the research constitutes a constraint, all relevant clinical data were recorded accurately and thoroughly in the hospital’s paper and electronic datasheets. Indeed, despite the uniqueness of this study in exploring the impact of inflammatory markers on mTESE outcomes, the limited number of participants could potentially affect the generalizability of the results. Therefore, future research involving larger patient cohorts would be beneficial to further validate and build upon our findings.

Overall, our study contributes to the growing body of literature on the prognostic factors associated with sperm retrieval in NOA. Testicular histopathology remains a significant predictor of successful sperm retrieval, highlighting the importance of DTB in guiding treatment decisions for patients with NOA. While age, reproductive hormone levels, inflammatory markers, and varicocele did not demonstrate significant predictive value in our study, further research is needed to fully understand their impact on sperm retrieval success in this population.

## 5. Conclusions

Our research indicates a discrepancy between the outcomes of DTB and TTB among a specific subset of patients with NOA. We found that DTB failed to identify sperm in approximately half of the men with NOA who later had sperm detected via TTB. Consequently, it may be prudent to forgo DTB prior to TTB, aligning with the guidelines provided by the European Association of Urology. Despite its limitations, DTB can still play an instrumental role in assessing spermatogenetic testicular damage and ruling out neoplasia. While it shows high positive predictive value for successful sperm retrieval in TTB, it falls short in predicting individual foci of active spermatogenesis in NOA patients’ testes. Notably, a negative result in DTB should not dissuade NOA patients from future TTB attempts or ICSI cycles. In essence, DTB’s function should be seen as advisory, providing valuable insight into the potential for sperm retrieval without offering definitive identification. Our study also found that common preoperative variables such as patient age and various hormonal and inflammatory biomarkers do not reliably predict successful sperm retrieval in NOA patients. Intriguingly, we discovered higher levels of inflammatory markers, namely NLR and PLR, in TTB-negative patients, hinting at a potential negative impact of systemic inflammation on sperm detection via mTESE. While our study provides valuable insights, further research with larger cohorts is essential to validate and expand upon our findings.

## Figures and Tables

**Figure 1 jpm-13-01362-f001:**
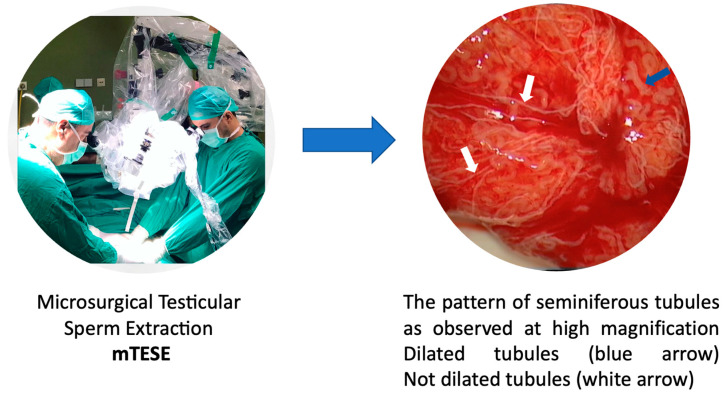
Identification of dilated tubuli during mTESE.

**Figure 2 jpm-13-01362-f002:**
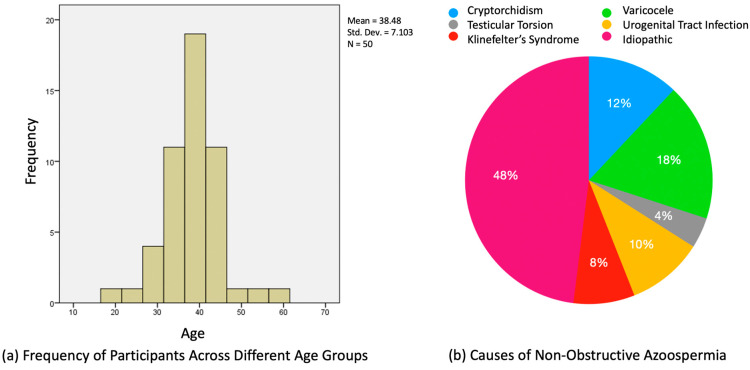
(**a**) Age distribution of men diagnosed with NOA; (**b**) pie chart representation of NOA etiology.

**Figure 3 jpm-13-01362-f003:**
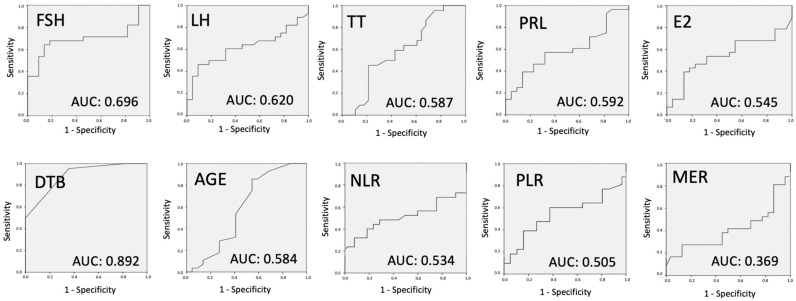
ROC curve analysis to evaluate the diagnostic accuracy of all variables individually. TTB: therapeutic testicular biopsy; FSH: follicle-stimulating hormone; LH: luteinizing hormone; TT: total testosterone; PRL: prolactin; E2: estradiol; DTB: diagnostic testicular biopsy; NLR: neutrophil-to-lymphocyte ratio; PLR: platelet-to-lymphocyte ratio; MER: monocyte-to-eosinophil ratio.

**Figure 4 jpm-13-01362-f004:**
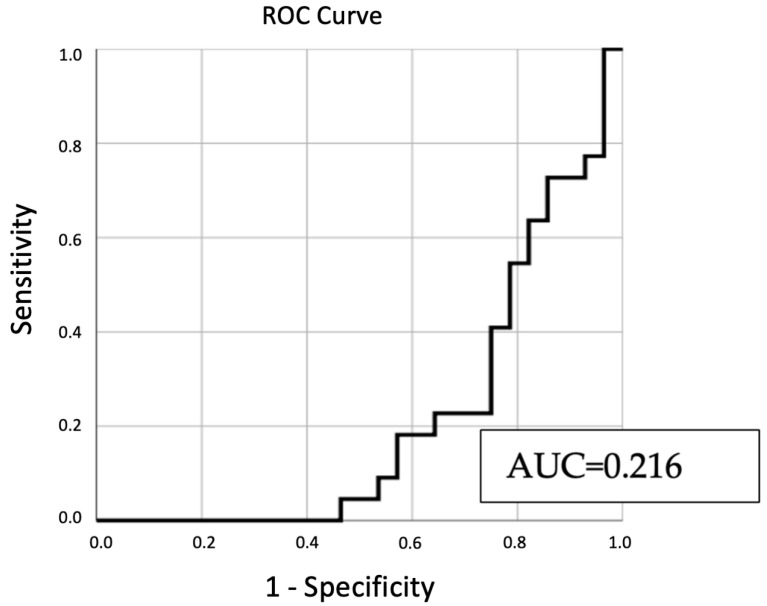
ROC curve analysis of predictive variables for sperm retrieval outcomes.

**Table 1 jpm-13-01362-t001:** Percentage of positive and negative therapeutic testicular biopsy according to the histological pattern.

Variable		TTB Negative	TTB Positive	Total	χ^2^	*p*-Value
DTB	HYPO	0 (0.0%)	11 (100%)	11	25.939	<0.05
	SCOS	13 (92.9%)	1 (7.1%)	14		
	MA	15 (60%)	10(40%)	25		
Total		28 (56%)	22 (44%)	50		

DTB: diagnostic testicular biopsy; TTB: therapeutic testicular biopsy; HYPO: hypospermatogenesis; SCOS: Sertoli cell-only syndrome; MA: maturation arrest.

**Table 2 jpm-13-01362-t002:** Preoperative peripheral hormonal serum values and blood biomarkers of systemic inflammation.

	Total (*n* = 50)		TTB-Negative (*n =* 28)		TTB-Positive (*n =* 22)		*t*-test	df	*p*-Value
Variable	Mean	±SD	Mean	±SD	Mean	±SD			
Age	38.5	7.1	39.5	4.9	37.2	9.1	1.036	48	0.308
FSH (1.3–19.3) IU/L	20.2	12.9	25.2	13.9	13.8	7.8	3.655	48	0.001
LH (1.2–8.6) IU/L	7.7	4.8	8.8	5.6	6.4	3.0	1.976	48	0.055
TT (1.8–7.8) ng/mL	2.9	1.1	2.8	1.2	3.1	0.9	−0.894	48	0.376
PRL (2.6–13.1) ng/mL	7.1	3.1	7.6	3.6	6.3	2.1	1.644	48	0.107
E2 (73–275) pmol/L	121.2	39.5	124.6	45.6	116.9	30.7	0.675	48	0.503
NLR	2.10	1.09	2.35	1.32	1.78	1.29	1.855	48	0.038
PLR	104.52	32.20	117.13	32.68	88.47	23.73	3.456	48	0.001
MER	3.80	3.44	3.76	2.97	3.86	4.04	0.100	48	0.459
TV	10.65	3.40	8.45	3.21	12.89	3.91	0.100	48	0.342

TTB: therapeutic testicular biopsy; FSH: follicle-stimulating hormone; LH: luteinizing hormone; TT: total testosterone; PRL: prolactin; E2: estradiol NLR: neutrophil-to-lymphocyte ratio; PLR: platelet-to-lymphocyte ratio; MER: monocyte-to-eosinophil ratio; TV: testicular volume.

**Table 3 jpm-13-01362-t003:** Multiple logistic regression analysis with successful sperm retrieval as the outcome variable.

Variable	Coefficient (β)	Standard Error (SE)	Wald’s Statistic	*p*-Value	Odds Ratio (Exp(β))	95% Confidence Interval (CI)
Age	−0.030	0.048	0.380	0.538	0.971	0.884–1.067
FSH	−0.110	0.057	3.772	0.052	0.895	0.801–1.001
LH	0.141	0.124	1.306	0.253	1.152	0.904–1.467
TT	0.323	0.404	0.641	0.424	1.382	0.626–3.051
PRL	0.022	0.152	0.020	0.886	1.022	0.758–1.378
E2	−0.007	0.009	0.584	0.445	0.993	0.976–1.011
NLR	0.268	0.585	0.210	0.647	1.307	0.415–4.115
PLR	0.002	0.016	0.017	0.897	1.002	0.972–1.033
MER	0.205	0.191	1.154	0.283	1.228	0.844–1.784
TV	−0.335	0.041	0.564	0.421	0.728	0.620–0.882
DTB	4.701	2.078	5.120	0.024	110.057	1.876–6457.484

TTB: therapeutic testicular biopsy; FSH: follicle-stimulating hormone; LH: luteinizing hormone; TT: total testosterone; PRL: prolactin; E2: estradiol NLR: neutrophil-to-lymphocyte ratio; PLR: platelet-to-lymphocyte ratio; MER: monocyte-to-eosinophil ratio; TV: testicular volume; DTB: diagnostic testicular biopsy.

**Table 4 jpm-13-01362-t004:** Analysis of variance (ANOVA) for predictors of TTB.

ANOVA ^a^					
Model	Sum of Squares	Df	Mean Square	F	Sig.
Regression	2.862	10	0.286	1.180	0.333 ^b^
Residual	9.458	39	0.243		
Total	12.320	49			

^a^ Dependent variable: TTB; ^b^ Predictors: (Constant), DTB, Age, NLR, E2, FSH, PRL, TT, MER, PLR, LH.

**Table 5 jpm-13-01362-t005:** Summary of regression model for predicting TTB Using multiple predictors.

Model Summary				
Model	R	R Square	Adjusted R Square	Std. Error of the Estimate
Regression	0.482 ^a^	0.232	0.035	0.492

^a^ Predictors: (Constant), DTB, Age, NLR, E2, FSH, PRL, TT, MER, PLR, LH.

**Table 6 jpm-13-01362-t006:** ROC curve analysis results for predicted probability of sperm retrieval.

Area Under the Curve				
Area	Std. Error ^a^	Asymptotic Sig. ^b^	Asymptotic 95% Confidence Interval	
			Lower Bound	Upper Bound
0.216	0.065	0.001	0.089	0.343

^a^ Under the nonparametric assumption; ^b^ Null hypothesis: true area = 0.5.

**Table 7 jpm-13-01362-t007:** Cohen’s kappa measure of agreement on predicted and observed sperm retrieval outcomes.

Symmetric Measures			
	Value	Asymptotic Standard Error ^a^	Approximate T ^b^
Measure of Agreement	Kappa	0.000	0.000
N of Valid Cases	50		

^a^ Not assuming the null hypothesis; ^b^ Using the asymptotic standard error assuming the null hypothesis.

**Table 8 jpm-13-01362-t008:** Sensitivity, specificity, PPV, NPV, and accuracy with 95% confidence interval of DTB in successfully finding sperm in TTB (sperm positive TTB).

Variable	Sensitivity (%) (95% CI)	Specificity (%) (95% CI)	PPV (%) (95% CI)	NPV (%) (95% CI)	Accuracy (%)
DTB	50 (28.22–71.78)	100 (87.66–100)	100	71.79 (62.63–79.45)	78 (64.04–88.47)

DTB: diagnostic testicular biopsy; TTB: therapeutic testicular biopsy; PPV: positive predictive value; NPV: negative predictive value.

## Data Availability

Not applicable.

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
