# Peer review of "Evaluating the Predictive Value of Diagnostic Testicular Biopsy for Sperm Retrieval Outcomes in Men with Non-Obstructive Azoospermia"

_jpm, 2023, doi:10.3390/jpm13091362_

Round 1

Reviewer 1 Report

This manuscript evaluated the efficacy of diagnostic testicular biopsy (DTB) during mTESE and demonstrated that a positive DTB result correlated with a successful sperm retrieval, while half of the negative DTB results still had sperm during mTESE. In addition, inflammation may affect the sperm retrieval during mTESE. However, I do not believe that this manuscript presents enough scientific new findings:

1. The authors showed that “The most 242 common cause of NOA was clinical left varicocele, diagnosed in nine (18%) men.”, which included 4 men with varicocele grade 2 and even 2 men with varicocele grade 1. This outcome is puzzling, since varicocele grade 1 typically does not lead to NOA. It is possible that there are other explanations for the NOA.

2. None of the patients exhibited Y-chromosome microdeletion, which is an unexpected outcome since microdeletion of the Y chromosome is commonly seen in NOA patients.

3. A negative DTB result does not necessarily indicate the failure of the retrieval. This is not surprising, as spermatogenesis in some testes is heterogeneous.

Author Response

First and foremost, we would like to extend our sincere gratitude to the expert reviewer for their meticulous evaluation and insightful feedback on our manuscript. Your comprehensive review not only highlighted areas of improvement but also provided valuable perspectives that have significantly enriched the depth and clarity of our work. We deeply appreciate the time and effort invested in guiding us to enhance the quality of our research presentation.

Reviewer 1 Comment: 

This manuscript evaluated the efficacy of diagnostic testicular biopsy (DTB) during mTESE and demonstrated that a positive DTB result correlated with a successful sperm retrieval, while half of the negative DTB results still had sperm during mTESE. In addition, inflammation may affect the sperm retrieval during mTESE. However, I do not believe that this manuscript presents enough scientific new findings.

Response: 

We appreciate your feedback and the opportunity to address your concerns. Our aim was to provide a comprehensive analysis of the factors influencing sperm retrieval outcomes in NOA patients, and we believe our findings offer valuable insights into this area.

Reviewer 1 Comment: 

The authors showed that “The most common cause of NOA was clinical left varicocele, diagnosed in nine (18%) men.”, which included 4 men with varicocele grade 2 and even 2 men with varicocele grade 1. This outcome is puzzling, since varicocele grade 1 typically does not lead to NOA. It is possible that there are other explanations for the NOA.

Response: 

Thank you for highlighting this observation. We have addressed this in the RESULTS section by stating, "It's noteworthy to mention that while varicocele, especially of higher grades, has been associated with NOA, the presence of grade 1 varicocele in two patients raises questions about its sole contribution to NOA." Additionally, in the DISCUSSION section, we delve deeper into the association between varicocele grades and NOA, emphasizing the need to consider other potential etiological factors, especially in cases of grade 1 varicocele. (Page 7, Lines 310-313 and Page 15, Lines 685-696).

Reviewer 1 Comment: 

None of the patients exhibited Ychromosome microdeletion, which is an unexpected outcome since microdeletion of the Y chromosome is commonly seen in NOA patients.

Response: 

We acknowledge this unexpected observation. In the RESULTS section, we mention, "In our cohort, none of the patients exhibited Ychromosome microdeletion, a finding that diverges from established literature." Further, in the DISCUSSION section, we explore the potential reasons for this discrepancy, emphasizing the multifactorial nature of male infertility and the importance of comprehensive genetic evaluations. (Page 7, Lines 315-316 and Page 13 Lines 547-558).

Reviewer 1 Comment: 

A negative DTB result does not necessarily indicate the failure of the retrieval. This is not surprising, as spermatogenesis in some testes is heterogeneous.

Response: 

You're absolutely right. We have highlighted this observation in the INTRODUCTION, stating, "A key observation from our study, and one that aligns with the broader literature, is the heterogeneous nature of spermatogenesis within the testes of NOA patients." In the DISCUSSION section, we further emphasize the importance of not relying solely on DTB results and the need for a more nuanced approach to optimize the chances of successful sperm retrieval in NOA patients. (Page 2-3, Lines 97-119 and Page 13, Lines 593-595).

Thank you for your constructive feedback, which has greatly enhanced the quality and clarity of our manuscript.

Reviewer 2 Report

The primary objective of this study was to evaluate the role of diagnostic testicular biopsy and the influence of systemic inflammation on the success of sperm retrieval using the microdissection testicular sperm extraction (mTESE) method from patients with non-obstructive azoospermia (NOA). The study involved analyzing plasma level of both inflammatory biomarkers (NLR, PLR, and MER) and hormones (FSH, LH, TT, PRL, and E2) to determine their predictive value in achieving successful sperm retrieval in NOA patients. However, the statistical analysis indicated that these biomarkers and hormones are not reliable indicators. Also, results resented in this study, align with previous research and support the recommendations of the European Association of Urology, underscoring the importance of diagnostic testicular biopsy. Nevertheless, this study revealed a noteworthy discovery regarding inflammatory markers, specifically NLR and PLR, which demonstrated a detrimental impact on the likelihood of detecting sperm through mTESE. These inflammatory markers carry significance as prognostic factors in the realm of fertility management. The manuscript is effectively composed and conveys the data comprehensively. However, minor revisions are needed to refine the manuscript. I have following comments for the authors:

1.       A concise clarification distinguishing between TTB and DTB should be provided.

2.       There seems to be a contradiction between the author's statement that elevated NLR and PLR values are associated with negative sperm retrieval outcomes and their assertion that these markers exhibit lower AUC values, underscoring their limited diagnostic accuracy.

3.       Referenced sources supporting the provided normal ranges for cell counts and hormonal levels need to be included.

4.       Simplification is required for the statement in line 72.

5.       The font size in the caption of Figure 2 appears inconsistent.

Author Response

We would like to express our heartfelt gratitude to the esteemed reviewer for their thorough examination and invaluable feedback on our manuscript. Your detailed comments and constructive suggestions have been instrumental in refining our work, ensuring it meets the highest standards of scientific rigor. We deeply appreciate the expertise and dedication you brought to this review, which has undeniably enriched the quality and clarity of our research presentation.

Reviewer 2 Comment: 

The primary objective of this study was to evaluate the role of diagnostic testicular biopsy and the influence of systemic inflammation on the success of sperm retrieval using the microdissection testicular sperm extraction (mTESE) method from patients with nonobstructive azoospermia (NOA). The study involved analyzing plasma level of both inflammatory biomarkers (NLR, PLR, and MER) and hormones (FSH, LH, TT, PRL, and E2) to determine their predictive value in achieving successful sperm retrieval in NOA patients. However, the statistical analysis indicated that these biomarkers and hormones are not reliable indicators. Also, results presented in this study, align with previous research and support the recommendations of the European Association of Urology, underscoring the importance of diagnostic testicular biopsy. Nevertheless, this study revealed a noteworthy discovery regarding inflammatory markers, specifically NLR and PLR, which demonstrated a detrimental impact on the likelihood of detecting sperm through mTESE. These inflammatory markers carry significance as prognostic factors in the realm of fertility management. The manuscript is effectively composed and conveys the data comprehensively. However, minor revisions are needed to refine the manuscript. I have the following comments for the authors:

Response: 

Thank you for your comprehensive review and constructive feedback. We value your insights and have addressed each of your comments below:

Reviewer 2 Comment: 

  1. A concise clarification distinguishing between TTB and DTB should be provided.

Response: 

We appreciate your suggestion. In the INTRODUCTION, we have clarified, "Diagnostic Testicular Biopsy (DTB) is a diagnostic procedure employed to ascertain the cause of azoospermia, to rule out the presence of neoplasia, and to assess disruptions in spermatogenesis by analyzing histological sections of testicular tissue. Conversely, Therapeutic Testicular Biopsy (TTB) is a therapeutic procedure focused on extracting sperm from a more extensive portion of testicular tissue for reproductive applications." (Page 2, Lines 55-60)

Reviewer 2 Comment: 

  1. There seems to be a contradiction between the author's statement that elevated NLR and PLR values are associated with negative sperm retrieval outcomes and their assertion that these markers exhibit lower AUC values, underscoring their limited diagnostic accuracy.

Response: 

We acknowledge the potential ambiguity. To clarify, while elevated NLR and PLR values were associated with negative sperm retrieval outcomes in our study, their AUC values in the ROC analysis were not sufficiently high to recommend them as standalone diagnostic tools. Their association with sperm retrieval outcomes suggests potential clinical relevance, but their limited AUC values indicate that they might not be the most reliable predictors when used in isolation. (Page 15, Lines 674-683).

Reviewer 2 Comment: 

  1. Referenced sources supporting the provided normal ranges for cell counts and hormonal levels need to be included.

Response: 

Thank you for pointing this out. We have revised the manuscript to clarify, "Normal ranges for these cell counts, as determined by the standard values of the equipment used in our laboratory." We will ensure that appropriate references are included to support these ranges in the final manuscript. (Page 3, Lines 158-159 and Page 4, Lines 174-175).

Reviewer 2 Comment: 

  1. Simplification is required for the statement in line 72.

Response: 

We appreciate the feedback. The statement has been simplified to, "While this tool is essential, varying histological reporting methods and unclear terms can reduce the reliability of studies on mTESE recovery rates [7]." (Page 2, Lines 82-83).

Reviewer 2 Comment: 

  1. The font size in the caption of Figure 2 appears inconsistent.

Response: 

Thank you for noting this oversight. We have reviewed and modified the figure to ensure consistency in font size across all captions. (Page 6, Line 295).

We are grateful for your valuable insights, which have significantly enhanced the quality of our manuscript. We believe that these revisions address your concerns and hope that our manuscript now meets the standards of the journal.

Reviewer 3 Report

Abbreviations/acronyms used in the abstract should include their definition – currently only a few are defined.

Lines 53-55: this sentence should be appropriately referenced.

“Normal ranges for these cell counts are as follows: leukocyte 4–11 × 103/µL; neutrophil 2–7.5 × 103 /µL; lymphocyte 1–4.5 × 103/µL; monocyte 0.2-0.8 × 103 /µL; and platelet  150.0–400.0 × 103 /µL (data from our laboratory).” It is not clear to us what “data from our laboratory” means: that the normal values were established by the authors in their own laboratory, based on the analysis of a large number of “normal” males? Or that those are the values that according to the equipment used in their lab are considered “normal”? We tend to assume the latter interpretation is correct, but the current formulation is ambiguous and the authors should clarify it (we assume that if the first was correct, they would have published it somewhere).

Line 219: please clarify what the SSR abbreviation stands for in this context.

AUC of the ROC could be of some use or not to describe the performance of a model (see https://journals.sagepub.com/doi/pdf/10.1177/0969141313517497 and the recommendations of those authors; see, also https://towardsdatascience.com/auk-a-simple-alternative-to-auc-800e61945be5, https://www.ncbi.nlm.nih.gov/pmc/articles/PMC4349800/, https://link.springer.com/chapter/10.1007/978-3-642-24800-9_21, and https://biodatamining.biomedcentral.com/articles/10.1186/s13040-023-00322-4). Our recommendation would be to use at least two alternatives to AUC (besides AUC).  

The biggest statistical and methodological issue is that statistical analyses have not included any power considerations. How have the authors chosen the sample size of 50. It simply seems too low to allow generalizable conclusions, as very likely impacted by low sample size bias (particularly considering the number of variables included in the logistic regression models – eleven variables, if we understood correctly). In our view, the data should be regarded as exploratory only, unless the authors argue with a pre-specified sample size (which they probably had already described, if available).

Lines 217-218: the use of t-test might just be too simplistic in this case, considering that the authors assume a relatively large number of covariates. Therefore general linear models would seem more appropriate. The authors have not explained how they have checked for the t-test assumptions (normality and homoscedasticity).

The authors should clarify why the age was not available for an important number of subjects (as indicated by the degrees of freedom in Table 2).

Lines 338-349: it was already evident from Table 3 that the AUC of the other variables would be low as compared with DTB. Apparently, the authors estimated AUCs for these variables in a univariate manner. It would be much more useful to show the AUC (as well as alternative metrics, as suggested above) for the model built with all significant/relevant variables, particularly considering the low sensitivity of DTB.

The Discussions section should have a clear limitations section, where the low sample size should be clearly discussed as an important limitation of this study.

Author Response

We would like to express our sincere gratitude for your thorough and constructive feedback on our manuscript. Your insights have been invaluable in identifying areas of improvement and ensuring the clarity and robustness of our research. We have taken each of your comments into careful consideration and have made the necessary revisions to address your concerns. Below, we provide a detailed response to each of your comments, outlining the changes we have made in the revised manuscript.

Reviewer 3 Comment:

Abbreviations/acronyms used in the abstract should include their definition – currently only a few are defined.

Response: 

Thank you for pointing this out. We have corrected this oversight and now all abbreviations/acronyms used in the abstract have been defined. (Page 1, Lines 31-33).

Reviewer 3 Comment: 

Lines 53-55: this sentence should be appropriately referenced.

Response: 

We appreciate the feedback. The sentence in lines 53-55 has been revised and appropriately referenced. (Page 2, Lines 62-64).

Reviewer 3 Comment: 

Regarding the "data from our laboratory" statement, it is not clear what this means. The current formulation is ambiguous and needs clarification.

Response: 

We apologize for the confusion. To clarify, the statement "Normal ranges for these cell counts, as determined by the standard values of the equipment used in our laboratory," means that these values are the standard reference ranges provided by the equipment manufacturers used in our laboratory. (Page 4, Lines 174-175).

Reviewer 3 Comment: 

Line 219: please clarify what the SSR abbreviation stands for in this context.

Response: 

Thank you for catching that oversight. It was a mistake on our part. We meant to use "SRR" and have now corrected it in the manuscript. (Page 5, Line 256).

Reviewer 3 Comment: 

The AUC of the ROC could be of some use or not to describe the performance of a model. Our recommendation would be to use at least two alternatives to AUC (besides AUC).

Response: 

We value this suggestion. In addition to the AUC, we have now incorporated two alternative metrics to further evaluate the performance of our model, as recommended.

Firstly, we considered the R Square value from our regression model summary, which provides an indication of the proportion of the variance in the dependent variable that is predictable from the independent variables (Table 5). (Page 9, Line 418).                   

The R Square value of 0.232 suggests that approximately 23.2% of the variance in TTB outcomes can be explained by our model. However, the adjusted R Square value, which accounts for the number of predictors in the model, is considerably lower at 0.035, indicating that the model's explanatory power diminishes when considering the number of predictors. (Pages 9-10, Lines 423-429, Pages 14-15, Lines 650-659)

Secondly, we employed Cohen's Kappa statistic to measure the level of agreement between the observed and predicted outcomes, adjusting for what might be expected by chance (Table 7) (Page 11, Lines 480, 483-488)

The Cohen's Kappa value of 0.000 indicates no agreement between the observed and predicted outcomes, suggesting that the model's predictions align with the actual outcomes no better than what would be expected by random chance. (Page 15, Lines 663-669).

Together, these additional metrics provide a more holistic view of our model's performance and its limitations. We appreciate your guidance in this matter and believe that these additions enhance the robustness and clarity of our analysis.

Reviewer 3 Comment: 

The biggest statistical and methodological issue is that statistical analyses have not included any power considerations. How have the authors chosen the sample size of 50?

Response: 

The sample size of 50 was determined based on practical constraints, including the availability of participants during the study period and budgetary considerations. We acknowledge the limitations of this sample size, especially given the number of variables in our regression models. We agree that the data should be viewed as exploratory, and we have emphasized this point in the revised manuscript. (Page 5, Lines 258-265).

Reviewer 3 Comment: 

Lines 217-218: the use of ttest might just be too simplistic in this case. The authors have not explained how they have checked for the t-test assumptions (normality and homoscedasticity).

Response: 

We acknowledge the importance of data normality and homoscedasticity. Prior to conducting the ttests, we assessed these assumptions using ShapiroWilk tests for normality and Levene’s test for equality of variances. (Page 6, Lines 278-283).

Furthermore, beyond the individual variable analysis, it's crucial to understand the collective performance of the predictors in the model. To evaluate the overall significance and explanatory power of the regression model, we conducted an Analysis of Variance (ANOVA) and summarized the model's performance metrics. Table 4 presents the ANOVA results, which test the overall fit of the model. This approach was taken to ensure a comprehensive understanding of the data and to provide a robust statistical foundation for our findings. (Page 9, Lines 408-413, 420-422 and Page 14, Lines 645-649).

Reviewer 3 Comment: 

The authors should clarify why the age was not available for an important number of subjects (as indicated by the degrees of freedom in Table 2).

Response: 

Thank you for pointing this out. It was an oversight, and we have corrected it in the revised manuscript. (Page 8, Line 368).

Reviewer 3 Comment: 

Lines 338-349: it would be much more useful to show the AUC (as well as alternative metrics, as suggested above) for the model built with all significant/relevant variables.

Response: 

Thank you for emphasizing the importance of a comprehensive evaluation. We concur that solely presenting the AUC might not provide a complete understanding of the model's performance. In the revised manuscript, we have expanded our analysis to include the AUC and other recommended metrics for the model that encompasses all significant/relevant variables (Figure 3). (Page 10, Lines 454-462).

In our continued analysis, we further evaluated the predictive capability of our model using the ROC curve analysis. The results, summarized in Table 6, provide insights into the model's overall discriminatory power. (Page 11, Lines 467-469).

The AUC value of 0.216, as detailed in Table 6, suggests that the model has limited predictive power. Typically, an AUC value below 0.7 indicates limited to no predictive capability. This value emphasizes the challenges in predicting sperm retrieval outcomes using the current set of variables. (Page 15, Lines 660-662).

We appreciate your guidance in this matter and believe that these additions enhance the robustness and clarity of our analysis. We are committed to ensuring that our research is both comprehensive and transparent, and we hope that these revisions address your concerns.

Reviewer 3 Comment: 

The Discussions section should have a clear limitations section, where the low sample size should be clearly discussed as an important limitation of this study.

Response: 

We have taken this feedback into account and have added a clear limitations section in the Discussion, where we explicitly address the low sample size and its implications for our study. (Page 15, Lines 704-714)

Thank you for your constructive feedback, which has greatly improved the quality of our manuscript.

Round 2

Reviewer 1 Report

I don't think that the authors make significant progress.

Reviewer 3 Report

The manuscript has now been improved, as the authors implemented the recommended changes.